# *Gymnema Sylvestre* Supplementation Restores Normoglycemia, Corrects Dyslipidemia, and Transcriptionally Modulates Pancreatic and Hepatic Gene Expression in Alloxan-Induced Hyperglycemic Rats

**DOI:** 10.3390/metabo13040516

**Published:** 2023-04-04

**Authors:** Humaira Muzaffar, Iqra Qamar, Muhammad Bashir, Farhat Jabeen, Shahzad Irfan, Haseeb Anwar

**Affiliations:** 1Department of Physiology, Govt. College University Faisalabad, Faisalabad 38000, Pakistan; 2Department of Zoology, Govt. College University Faisalabad, Faisalabad 38000, Pakistan

**Keywords:** hyperglycemia, dyslipidemia, oxidative stress, *Gymnema sylvestre*

## Abstract

*Gymnema sylvestre* is traditionally used as an herbal remedy for diabetes. The effect of *Gymnema sylvestre* supplementation on beta cell and hepatic activity was explored in an alloxan-induced hyperglycemic adult rat. Animals were made hyperglycemic via a single inj. (i.p) of Alloxan. *Gymnema sylvestre* was supplemented in diet @250 mg/kg and 500 mg/kg b.w. Animals were sacrificed, and blood and tissues (pancreas and liver) were collected for biochemical, expression, and histological analysis. *Gymnema sylvestre* significantly reduced blood glucose levels with a subsequent increase in plasma insulin levels in a dosage-dependent manner. Total oxidant status (TOS), malondialdehyde, LDL, VLDL, ALT, AST, triglyceride, total cholesterol, and total protein levels were reduced significantly. Significantly raised paraoxonase, arylesterase, albumin, and HDL levels were also observed in *Gymnema sylvestre* treated hyperglycemic rats. Increased mRNA expression of *Ins-1, Ins-2, Gck, Pdx1, Mafa,* and *Pax6* was observed, while decreased expression of *Cat, Sod1*, *Nrf2*, and *NF-kB* was observed in the pancreas. However, increased mRNA expression of *Gck, Irs1, SREBP1c*, *and Foxk1* and decreased expression of *Irs2*, *ChREBP, Foxo1, and FoxA2* were observed in the liver. The current study indicates the potent effect of *Gymnema sylvestre* on the transcription modulation of the insulin gene in the alloxan-induced hyperglycemic rat model. Enhanced plasma insulin levels further help to improve hyperglycemia-induced dyslipidemia through transcriptional modulation of hepatocytes.

## 1. Introduction

Diabetes is a global metabolic disorder projected at a global incidence of 26.6 million, with a prevalence of 570.9 million in 2025 [1]. An abnormally high plasma level of glucose is the clinical feature of diabetes and results in abnormal plasma lipid concentrations with conditions such as hypercholesterolemia and dyslipidemia [2]. Abnormal levels of hepatic enzymes also known as liver function tests (LFTs) are also associated with dyslipidemia observed in diabetes mellitus [3]. Hyperglycemia also provokes the formation of cellular reactive oxygen species (ROS), resulting in secondary complications during diabetes mellitus [4,5,6]. Superoxide dismutase (SOD) and malondialdehyde (MDA) are vital antioxidant enzymes and their inactivity favors the progression of diabetes mellitus [7,8,9,10]. The hepatic metabolism of glucose and lipids by the hepatocytes is regulated by insulin. Thus, the liver primarily relies on insulin signaling to maintain metabolic homeostasis. Abnormal insulin levels, as well as ineffective insulin signaling observed in diabetes mellitus, limit the ability of the liver to maintain metabolic homeostasis resulting in hyperglycemia and dyslipidemia [11,12].

*Gymnema sylvestre* (GS), a woody climbing plant from the Asclepiadaceae family, has been used as traditional folk medicine. Its leaves contain several bioactive constituents, including gymnemagenol, gymnemic acids, gymnomosides, gurmarin, and gymnemanol [13,14,15,16]. These bioactive constituents have been attributed to impart in vitro antidiabetic properties, mainly through insulin release via the modulation of incretins [17,18]. The antioxidant ability of GS is due to the presence of many bioactive components in its leaves. These bioactive components are oleanane-type triterpenoid saponins, and are known as anthraquinones, acidic glycosides, gymnemic acids alkaloids, and their products [18,19]. The in vitro hepatoprotective activity of GS has also been reported previously [20,21,22]. It has been reported that GS stimulates insulin release in vitro by increasing the membrane permeability of pancreatic beta cells [23,24], and GS treatment has been reported previously to modulate incretin activity, thereby triggering insulin release [25].

In the present study, the antidiabetic and anti-hypercholesterolemic properties of GS were re-examined in the context of transcriptional modulation of critical genes involved in the regulation of insulin, insulin signaling, carbohydrate, and lipid metabolism in a hyperglycemic rat model. Additionally, the role of GS as an efficient hepato-protecting agent was also examined through its possible modulation of antioxidant response elements.

## 2. Experimental Design

The study design was rigorously discussed and formally approved by the university biosafety and bioethical boards per guidelines. All experimental protocols, including animal handling, were approved by the Institutional Animal Care and Ethical Committee, Government College University Faisalabad (GCUF/ERC/18/037). In total, 64 male Albino Wister rats weighing 230–250 g were obtained from the Experimental Animal Breeding Station, Department of Physiology, Govt. College University Faisalabad, Pakistan. Animals were acclimatized for two weeks in optimum conditions at the facility to minimize stress levels. A chow maintenance diet (CMD) and water ad libitum were provided to the rats. All healthy recruited rats were randomly placed into four groups, as shown in Table 1.

The hyperglycemic condition was induced in the rats by using Alloxan. Alloxan is a β-cytotoxin that induces pancreatic β-cell damage, thus causing chemical diabetes [26]. A single intraperitoneal injection of Alloxan (Sigma-Aldrich Co. St. Louis, MI, USA) at a dose of 120 mg/kg b.w. was used to induce hyperglycemia in adult rats. Blood glucose levels were measured by the tail-tipping method using a glucometer. The rats with blood glucose levels exceeding 200 mg/dL were considered diabetic. *Gymnema sylvestre* (GS) was obtained from a local botanical garden and identified by the botanist from the Department of Botany, Government College University Faisalabad. After crushing, the leaves were dried and powdered mechanically using a mortar and pestle. Both treatment groups (III and IV) were fed with CMD mixed with powdered GS for 21 days [Table 1]. On the 21st day of the treatment, the rats from all groups were decapitated. The whole blood of each rat was collected in separate 5 mL vials of blood vacutainer. Serum was extracted from these vials by centrifugation at 3000× *g* for 10 min and stored at −20 °C until further usage. Pancreas and liver tissue were collected, and part of the tissues was immediately stored in RNAlater^©^ for expression analysis, and part of the tissue was fixed in 10% formalin for pathological examination.

### 2.1. Blood Glucose and Plasma Insulin Assessment

The plasma glucose levels were measured using commercially available glucose strips and a glucometer (Accu-Chek^®^ Instant Blood glucose monitor by Roche). The competitive ELISA was employed to measure the plasma levels of insulin. A commercially available ELISA kit was used. Specifically, the RayBio^®^ Rat Insulin ELISA kit (CODE: ELR-Insulin-1) was utilized with the detection range and sensitivity of 5 uIU/Ml–300 uIU/mL and 5 uIU/mL, respectively.

### 2.2. Total Oxidant Status (TOS)

The method described by Nisar et al. [27] was used to assess the total oxidant status (TOS) of the serum samples. The TOS in samples was examined with H_2_O_2_ standards (6.25, 3.12, 1.56, 0.78, and 0.39 μmol/L). The lowest limit of detection of the assay was 0.13 μmol H_2_O_2_ equal·L^−1^, alongx< 3% precision and the linearity until 200 μmol H_2_O_2_ equal·L^−1^. The coefficient of variation of intra-assay was below 10%.

### 2.3. Paraoxonase Activity

The enzyme’s activity was examined by consuming 2 mmol·L^−1^ paraoxonase, which acts as a substrate for the assay. The previously described method by Nisar et al. [27] was employed to measure the enzymatic activity. The lowest limit of detection in the assay was 80–100 U·min^−1^·L^−1^. The initial rate and/or sensitivity of the hydrolysis were stable for up to 5 min, while the coefficient of variance for intra-assay was < 10%.

### 2.4. Arylesterase Activity

Lower serum arylesterase activity is associated with abnormal lipid metabolism by favoring lipoprotein oxidation. Per minute activity of arylesterase was measured by a method which is previously described by Nisar et al. [27]. The lowest limit of detection in this assay was 40–55 kU·min−1·L^−1^. The coefficient of variance in intra-assay was <7%. The hydrolysis rate was steady up to 5 min after beginning hydrolysis.

### 2.5. Serum Lipid Profile

Triglycerides (TG), Total Cholesterol (TC), and High-Density Lipoprotein-Cholesterol (HDL-Chol) levels were measured by commercially available kits from Crescent Diagnostic Systems, Jeddah, KSA (Cat#: CS611-4, Cat#: CS603-10, Cat#: CS607-2, respectively). The lowest limit of detection for TG and TC assays were 2 mg/dL and 3 mg/dL, respectively. However, LDL- Cholesterol (LDL-Chol) and Very low-Density Lipoprotein- Cholesterol (VLDL-Chol) were measured by specific calculations (Friedwald equation: Total cholesterol − HDL cholesterol = LDL cholesterol).

### 2.6. Serum Liver Enzymes

A commercially available liquiform method kit supplied by Randox Laboratories Ltd. (BT29 4QY; Crumlin, County Antrim, UK) was used to measure the aspartate aminotransferase (AST) and alanine aminotransferase (ALT) concentrations in the serum. The detection range of the AST kit was 7.20–1039 U/L. For ALT, the measuring range was 9.7–603 U/L. Linearity was maintained when the absorbance of the sample was <0.5 and the CV was <10%. Total protein and albumin were also measured using colorimetric assay through a commercially available kits by Sigma-Aldrich St. Louis, MI, USA (Total protein kit Cat#: TP0100, BCP Albumin assay kit Cat#: MAK125). The sensitivity of the kit for total protein was 16.5 ug/mL, with a maximum detection limit of 1000 ug/mL. Intra and inter-assay CV was 2.2% and 4.5%, respectively.

### 2.7. Real-Time qPCR Analysis

The qPCR method was used to detect the expression of related mRNA levels in the pancreatic and liver tissue. Total RNA was extracted using the TRIzol reagent (Invitrogen, Waltham, MA, USA) and evaluated for concentration and purity through Nanodrop 2000 spectrophotometer (Thermo Fisher Scientific, Waltham, MA, USA). The total isolated mRNA was reverse-transcribed to cDNA using the RevertAid cDNA synthesis kit (Thermo Fisher Scientific, Waltham, MA, USA) according to the manufacturer’s manual. Real-time qPCR (RT-qPCR) was carried out on the iQ5 Bio-RAD machine using Maxima SYBR Green/ROX qRT-PCR Master Mix (Thermo Fisher Scientific, Waltham, MA, USA). Expression profiles of *Ins-1, Ins-2, Gck, Pdx1, MafA, Pax6, Cat, Sod1, Nrf2,* and *NF-kB* were observed in the pancreas. Whereas for the liver, the expression profile of the following genes was observed: *Irs1, Irs2, Gck, Sod1, Sod2, Cat, ChREBP, SREBP1c, Nrf2, NF-kb, FoxA2, FoxO1,* and *FoxK1*. The *β*-actin gene was used as a housekeeping/reference gene. The PCR was performed for 15 s, annealing for 25 s at 52 °C, and the extension time was 20 s at 72 °C. Specific primer sequences were used to amplify genes (Appendix A).

### 2.8. Histological Analysis

Pancreas and liver samples were subjected to H&E staining for histological examination to microscopically examine the size of the pancreatic islets of Langerhans and the hepatic sinusoids in liver parenchyma in different groups.

### 2.9. Statistical Analysis

All results were expressed as mean ± SEM. One-way ANOVA was used to analyze data using SPSS statistical tool package, and Tukey’s test was applied post hoc for comparing means among different groups. *p* ≤ 0.05 was considered statistically significant.

## 3. Results

### 3.1. GS Reduces Glucose Concentration and Increases Plasma Insulin Levels in a Dosage-Dependent Manner

The glucose levels significantly declined over time in the treatment groups as compared to the positive control (Figure 1A). The dose-dependent effect was also exhibited in the treatment groups during the treatment. The glucose levels in the positive control group displayed a significant rise after six days as compared to the negative control group. A significant drop in the plasma insulin level were observed in the PC group as compared to the NC group (Figure 1B). The GS treatment groups exhibited a dose dependent increase in the plasma insulin levels. The increase in the GS treated groups was found to be statistically significant as compared to the PC group.

### 3.2. GS Reduces Oxidative Stress in a Dosage-Dependent Manner

The TOS levels were significantly high in the positive control group in comparison with the negative control group (Figure 2A). A significant dose-dependent decline was observed in TOS values in both treatment groups. The paraoxonase levels exhibited a substantial decrease in the positive control group compared to the negative control group (Figure 2B). GS treatment groups showed a dose-dependent significant increase in the paraoxonase activity compared to the negative control group. Interestingly, the role of GS in mediating superoxide dismutase (SOD) levels was not statistically significant compared to the negative control group.

### 3.3. GS Improves Hepatic Response toward Oxidative Stress

Arylesterase activity was significantly reduced in the positive control group compared to the negative control group (Figure 3A). GS treatment greatly enhanced the arylesterase levels in a dose-dependent manner. Malondialdehyde (MDA) levels increased in the positive control group compared to the negative control group (Figure 3B). The difference was statistically non-significant. GS treatment lowered the malondialdehyde levels in a dose-dependent manner.

Aspartate aminotransferase (AST) and alanine aminotransferase (ALT) levels were markedly increased in the positive group as compared to the negative control group (Figure 4A,B). A dose-dependent decline was observed only for AST in the treatment groups as compared to the positive group. The higher dose of GSwas found to lower the AST levels, which were comparable with the negative control group levels.

### 3.4. GS Treatment Corrects Dyslipidemia

High-density lipoprotein (HDL) and serum albumin levels were found to be reduced profoundly in the positive control group as compared to the negative control group (Figure 5A,B). A significant (*p* ≤ 0.05) and dose-dependent rise was observed for HDL and albumin levels in the treatment groups as compared to the positive control group. Low-density lipoprotein (LDL), very low-density lipoprotein (VLDL), triglycerides, total cholesterol, and total protein levels were higher in the positive control group as compared to the negative control group (Figure 5C–G). The treatment groups exhibited a significant (*p* ≤ 0.05) decline in the aforementioned parameters as compared to the positive control group. The decrease observed in the treatment groups was also found to be treatment dose-dependent.

### 3.5. GS Treatment Significantly Altered Transcriptional Profile in the Pancreas and Liver

The expression profile for the specific genes in the pancreas and liver show significant variation among different treatment groups. In the pancreas, beta cell-specific genes, such as *Ins1* and *Ins2*, exhibit a significant decline in the positive control (PC) group followed by a gradual dose-dependent significant increase in the treatment groups (GS1, GS2) (Figure 6A). A similar pattern of mRNA expression was observed for the glucokinase gene (*Gck*) and different transcription factor genes, including pancreatic and duodenal homeobox 1 (*Pdx1),* MAF BZIP transcription factor A *(MafA),* and paired box protein 6 (*Pax6*) (Figure 6B). Antioxidant enzyme genes exhibit the same pattern of mRNA expression, where the positive control (PC) group depicts a significant decrease in the mRNA expression of catalase (*Cat*), superoxide dismutase 2 (*Sod2/MnSOD*), nuclear factor erythroid 2-related factor 2 (*Nrf2*). This significant increase was followed by a gradual decline in the treatment groups GS1 and GS2. However, the nuclear factor kappa B (*Nf-kB)* depicts higher mRNA expression in the positive control group, followed by a significant decrease in the treatment groups (GS1, GS2) (Figure 6C).

In the liver tissue, the genes related to the insulin signaling in the hepatocytes, such as insulin receptor substrates 1 and 2 (*Irs1, Irs2*), and the glucokinase gene (*GcK*), exhibit a distinct pattern of expression. *Irs1* and *Gck* mRNA expression declined significantly in the PC group, whereas the *Irs2* mRNA expression displayed a significant increase. Followed by a gradual and significant increase in the expression of of *Irs1* and *Gck* while a decrease in the *Irs2* mRNA expression was observed in the treatment groups GS1 and GS2 (Figure 7A). Superoxide dismutase enzyme genes (*Sod1, Sod2*), along with the catalase (*Cat*) gene, exhibit a peculiar pattern of mRNA expression where the PC group displays a significant decrease followed by a gradual increase in both treatment groups GS1 and GS2 (Figure 7B). The mRNA expression pattern of carbohydrate response element binding protein (*ChREBP*), a glucose-sensitive transcription factor, and sterol regulatory element binding protein 1c (*SREBP1c*), an insulin-sensitive transcription factor, displayed opposing expression patterns. The PC group exhibits a significant increase in the mRNA expression of *ChREBP* and a significant decline in the mRNA expression of *SREBP1c*. Followed by a gradual dose-dependent decline in the case of *ChREBP* and a dose-dependent rise in the case of *SREBP1c* mRNA expression in both treatment groups (Figure 7C). The mRNA expression of the insulin-sensitive forkhead family of transcription factors, including *FoxA2*, *Foxo1*, and *Foxk1*, displays a distinct pattern. *FoxA2* and *FoxO1* showed a similar expression pattern, and the PC group exhibited a significant increase followed by a gradual and significant decline in both treatment groups (Figure 7D). *Foxk1* mRNA expression displayed a significant decline in the PC group, followed by a gradual increase in the treatment groups (Figure 7D).

### 3.6. GS Treatment Modulates Pancreatic Islets and Hepatic Sinusoidal Space

The microscopic analysis of the pancreatic tissue reveals a decrease in the islet diameter and the total cell numbers in the positive control group (PC) compared to the negative control group (NC) (Figure 8A,B). The islets exhibit an increase in their diameter and total cell numbers in the treatment groups (GS1, GS2) as compared to the positive control group (PC) (Figure 8C,D)

The microscopic analysis of the liver parenchyma in different treatment groups revealed increased sinusoidal and perisinusoidal space in the positive control group (PC) as compared to the negative control group (NC) (Figure 9A,B). Both treatment groups exhibit decreased sinusoidal and perisinusoidal space, but in treatment group 1 (GS1) the decrement is absent near the central vein, while in the treatment group 2 (GS2) a more uniform reduction in the sinusoidal space is observed, including the area surrounding the central vein (Figure 9C,D).

## 4. Discussion

The present study examined the effect of GS-dried crushed leaves on different metabolic parameters in a hyperglycemic rat model. In the current study, we report an in vivo effect of GS treatment on plasma insulin levels and an increase in insulin gene transcription. We speculate that the initial rise in the plasma insulin levels might be due to the change in the membrane permeability of beta cells induced by incretins, resulting in insulin secretion and not synthesis. Secondly, the incretins are also responsible for the transcriptional activation of *Pdx1* [28,29,30]. Incretins, along with insulin, induced an increased expression of *Pdx1*, as insulin is also responsible for increased transcriptional activity of *Pdx1* [31]. Increased *Pdx1* transcription resulted in further enhancement of *Ins-1* expression and subsequent insulin synthesis (Figure 6A,B). This is depicted by a gradual increase in the insulin concentration in the treatment groups (Figure 1B).

Significant antioxidant activity of GS supplementation was observed in the present study in terms of the plasma levels of total oxidant status and hepatic enzymes such as paraoxonase, arylesterase, AST, and ALT. Paraoxonase and arylesterase are associated with high-density lipoproteins (HDLs) and inhibit the oxidation of low-density lipoproteins (LDL) [32,33]. Hyperglycemia is associated with elevated levels of AST and ALT [34]. Plasma albumin levels were raised in both treatment groups as plasma albumin levels are inversely associated with insulin reserves [35]. Superoxide dismutase (*Sod1, Sod2*) and catalase (*Cat*) are potent antioxidant enzymes and regulate the levels of reactive oxygen species (*ROS*) generated through hyperglycemia [36,37].

Nuclear factor erythroid 2-related factor (*Nrf2*) is a transcription factor that mediates the redox homeostatic gene regulatory network, and regulates the expression of *Sod1, Sod2,* and *Cat* expression and helps restore redox homeostasis [38]. *NF-kB* is an inflammation-sensitive transcription factor that initiates the cytokine response during inflammation [39]. A significant decrease in the mRNA expression of antioxidant enzyme genes such as *Sod1, Sod2, Cat,* and *Nrf2* was observed in the positive control group and followed by a gradual and significant increase in GS-treatment groups (Figure 6C and Figure 7B,C). Malondialdehyde (MDA) is a marker of lipid peroxidation and is associated with hyperglycemic-induced dyslipidemia [40]. MDA levels in the positive control group were increased non-significantly and followed by a reduction in both treatment groups.

Insulin receptor substrates (*Irs1, Irs2*) in hepatocytes are activated by insulin receptors [41]. Insulin receptor proteins are also responsible for the transcriptional upregulation of the hepatic glucokinase (*Gck*) gene and subsequent carbohydrate metabolism in the hepatocytes [42]. Irs1 has a constant hepatic expression, whereas *Irs2* is only expressed during fasting as *Irs2* is inhibited by insulin [43,44,45]. Hepatic glucokinase (*Gck*) expression also depends on feeding and fasted state and is primarily upregulated by insulin and downregulated by glucagon [46]. In the present study, the hepatic expression of *Irs1* and *Gck* was reduced, and that of *Irs2* was increased in the positive control group because of low plasma insulin levels (Figure 7A). Both treatment groups exhibited a marked increase in *Irs1* and *Gck* and a decline in *Irs2* levels, owing to increased plasma insulin levels (Figure 1B).

Hepatic expression of the insulin-sensitive forkhead family of transcription factors (*FOXOs*) has been shown to increase during fasting due to a lack of insulin [47]. *FoxO1* is involved in the hepatic production of glucose and *VLDL* in the absence of insulin, as increased hepatic insulin signaling has been shown to inhibit *FoxO1* and thereby reduce the release of glucose and *VLDL* from the liver by restraining microsomal triglyceride transfer protein (*MTP*) and apolipoprotein B (*ApoB*) expression [48,49,50]. *FoxA2* also stimulates hepatic expression of *MTB*, thereby enhancing *VLDL* release from the liver [51]. *FoxA2* is deactivated/phosphorylated by insulin, resulting in the decreased production of *MTB* and *VLDL* from the liver [52]. Recently *Foxk1* has been shown to be positively regulated by insulin, and results in the enhancement of glucose metabolism by hepatocytes [53]. In the present study, we speculate that the primary pathway through which GS supplementation improves dyslipidemia in hyperglycemic rats is mainly through enhanced hepatic signaling of insulin depicted by the increased transcriptional activity of *Irs1* and *Irs2* genes (Figure 7A).

The GS supplementation in the current study significantly enhanced sterol regulatory binding protein 1c (*SREBP1c*) mRNA levels in the liver. We speculate that increased plasma insulin levels and enhanced hepatic insulin signaling via Irs1 and Irs2 are responsible for increased *SREBP1c* expression in the GS-treated groups. As *SREBP1c* activation requires high physiological levels of insulin, activation of *SREBP1c* enhances the expression of key hepatic enzymes involved in fatty acid synthesis [54,55]. The GS supplementation also significantly reduced carbohydrate response element binding protein *(ChREBP)* mRNA expression in the liver, primarily by increased hepatic Gck activity, resulting in the reduction in plasma glucose levels. High glucose levels (hyperglycemia) and increased glucose metabolites have been shown to activate hepatic expression of *ChREBP,* resulting in the upregulation of hepatic lipogenic enzymes and favoring increased hepatic lipogenesis [56,57,58].

In conclusion, the present study suggests that *Gymnema sylvestre* leaves, in their crude form, can act as antioxidative, hepatoprotective, and hypocholesterolemic agents in an experimentally induced diabetic rat model.

## Figures and Tables

**Figure 1 metabolites-13-00516-f001:**
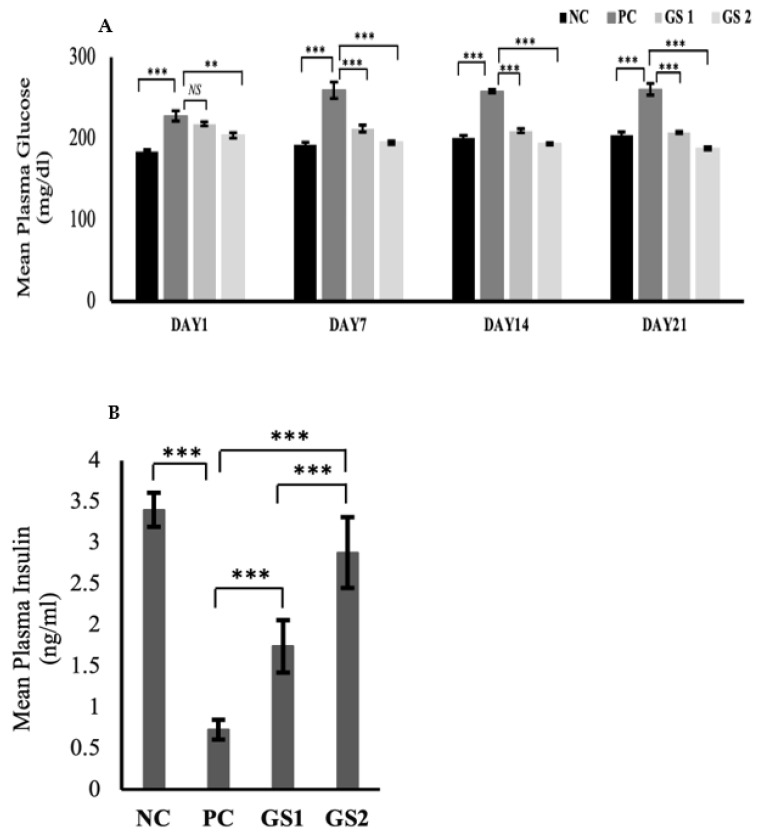
Mean values of serum glucose (mg/dl) (**A**) and plasma insulin (ng/mL) (**B**) levels in different treatment groups. Negative control (NC), Positive control (PC), *Gymnema sylvestre* 250 mg (GS1), *Gymnema sylvestre* 500 mg (GS2). *NS*: non-significant, (******): *p* ≥ 0.01, (*******): *p* ≥ 0.001.

**Figure 2 metabolites-13-00516-f002:**
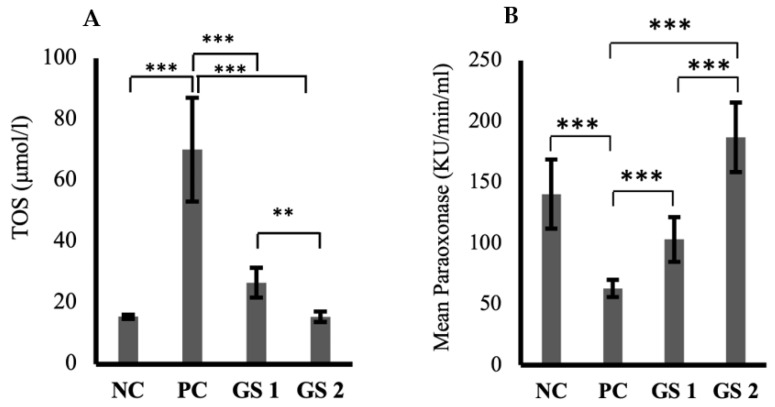
Mean values of total oxidant status (TOS, µmol/l) (**A**) and paraoxonase activity (KU/min/mL) (**B**) in different treatment groups. Negative control (NC), positive control (PC), *Gymnema sylvestre* 250 mg (GS1), *Gymnema sylvestre* 500 mg (GS2). (******): *p* ≥ 0.01, (*******): *p* ≥ 0.001.

**Figure 3 metabolites-13-00516-f003:**
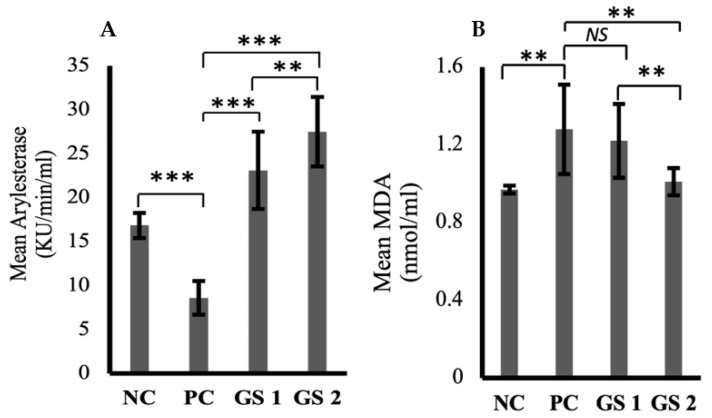
Mean values of arylesterase (Ku/min/mL) (**A**) and melanodialdehyde (MDA, nmol/mL) (**B**) activity in different treatment groups. Negative control (NC), Positive control (PC), *Gymnema sylvestre* 250 mg (GS1), *Gymnema sylvestre* 500 mg (GS2). *NS*: non-significant, (******): *p* ≥ 0.01, (*******): *p* ≥ 0.001.

**Figure 4 metabolites-13-00516-f004:**
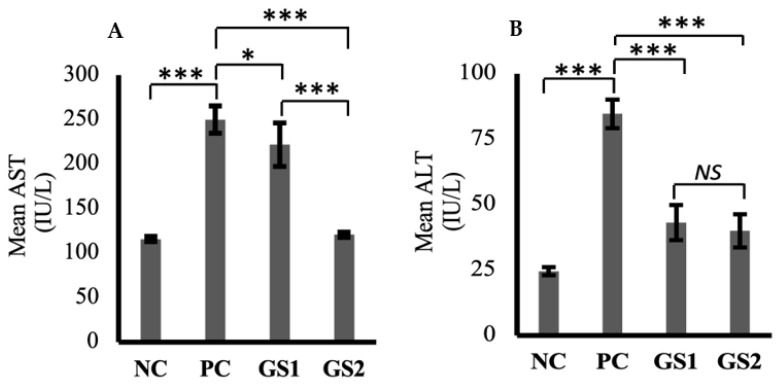
Mean values of aspartate aminotransferase (AST, IU/L) (**A**) and alanine aminotransferase (ALT, IU/L) (**B**) in different treatment groups. Negative control (NC), positive control (PC), *Gymnema sylvestre* 250 mg (GS1), *Gymnema sylvestre* 500 mg (GS2). *NS*: non-significant, (*****): *p* ≥ 0.05, (*******): *p* ≥ 0.001.

**Figure 5 metabolites-13-00516-f005:**
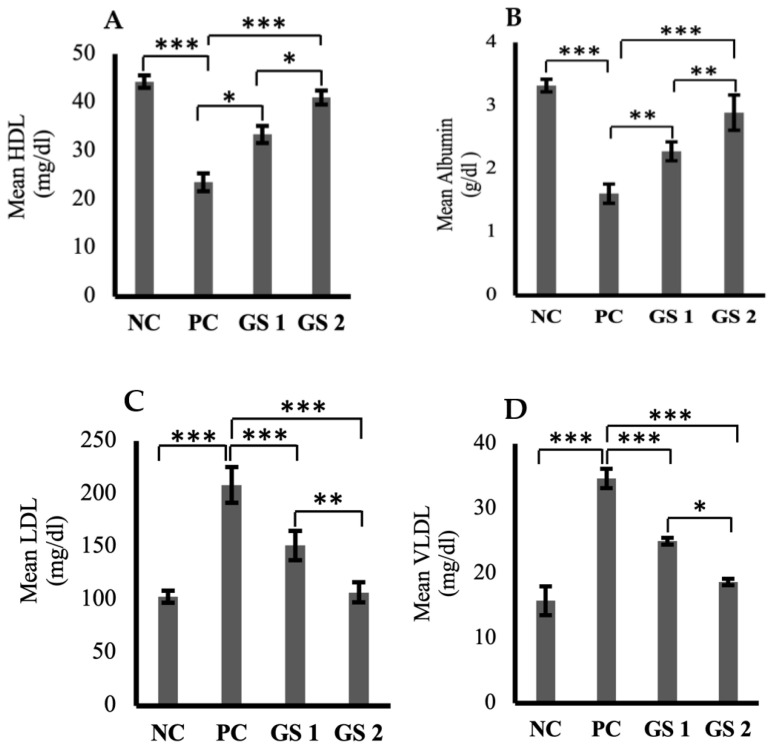
Mean plasma levels of high-density lipoprotein (HDL, mg/dl) (**A**), albumin (ALB, µg/mL) (**B**), low-density lipoproteins (LDL, mg/dl) (**C**), very low-density lipoprotein (VLDL, mg/dl) (**D**), triglycerides (TG, mmol/mL) (**E**), total cholesterol (TC, mg/dl) (**F**), total protein (TP, g/dl) (**G**) in different groups. Negative control (NC), positive control (PC), *Gymnema sylvestre* 250 mg (GS1), *Gymnema sylvestre* 500 mg (GS2). *NS*: non-significant, (*****): *p* ≥ 0.05, (******): *p* ≥ 0.01, (*******): *p* ≥ 0.001.

**Figure 6 metabolites-13-00516-f006:**
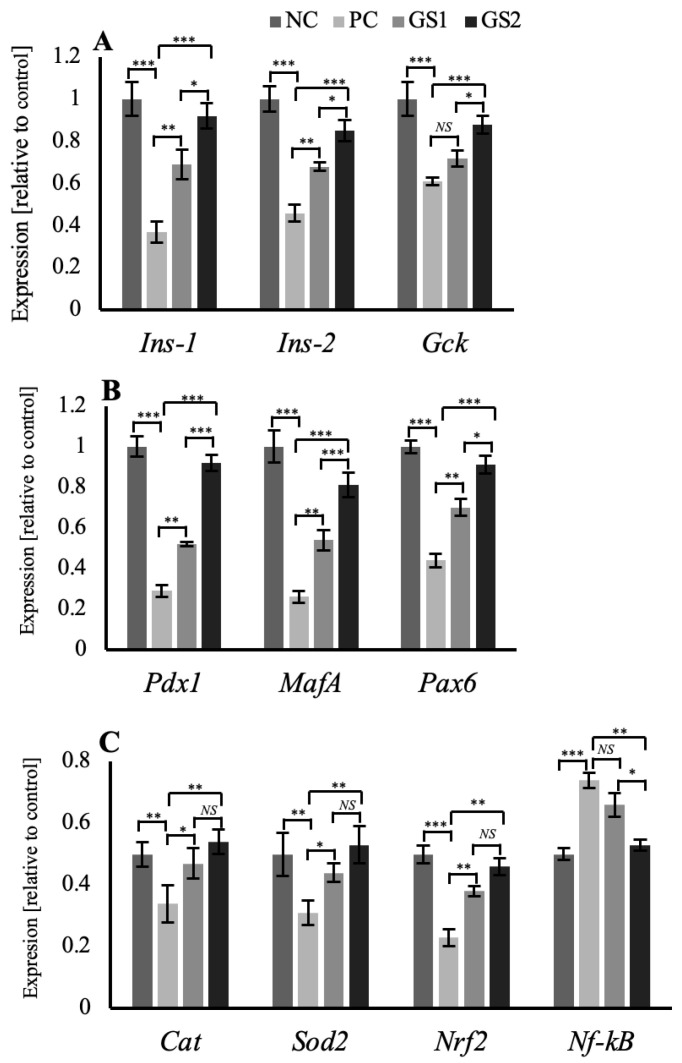
Mean values of mRNA expression in the pancreas of different treatment groups. (**A**) *Ins1, Ins2*, and *Gck.* (**B**) *Pdx1, MafA,* and *Pax6*. (**C**) *Cat, Sod2, Nrf2,* and *Nf-kB*. Negative control (NC), positive control (PC), *Gymnema sylvestre* 250 mg (GS1), *Gymnema sylvestre* 500 mg (GS2). *NS*: non-significant, (*****): *p* ≥ 0.05, (******): *p* ≥ 0.01, (*******): *p* ≥ 0.001.

**Figure 7 metabolites-13-00516-f007:**
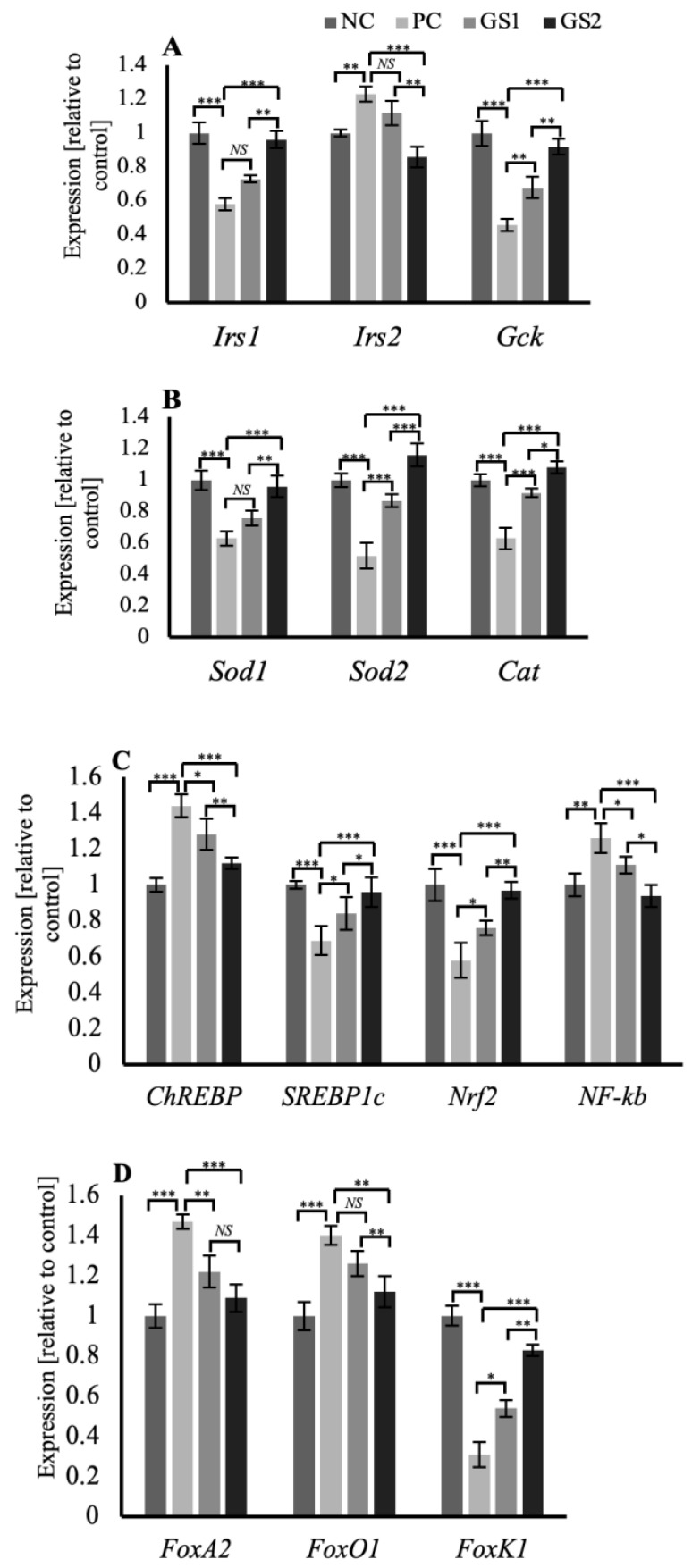
Mean values of mRNA expression of different genes in the liver of different treatment groups. (**A**) *Irs1, Irs2*, and *Gck* (**B**) *Sod1, Sod2,* and *Cat*. (**C**) *ChREBP, SREBP1c, Nrf2,* and *Nf-kB*. (**D**) *FoxA2, FoxO1,* and *FoxK1*. Negative control (NC), positive control (PC), *Gymnema sylvestre* 250 mg (GS1), *Gymnema sylvestre* 500 mg (GS2). *NS*: non-significant, (*****): *p* ≥ 0.05, (******): *p* ≥ 0.01, (*******): *p* ≥ 0.001.

**Figure 8 metabolites-13-00516-f008:**
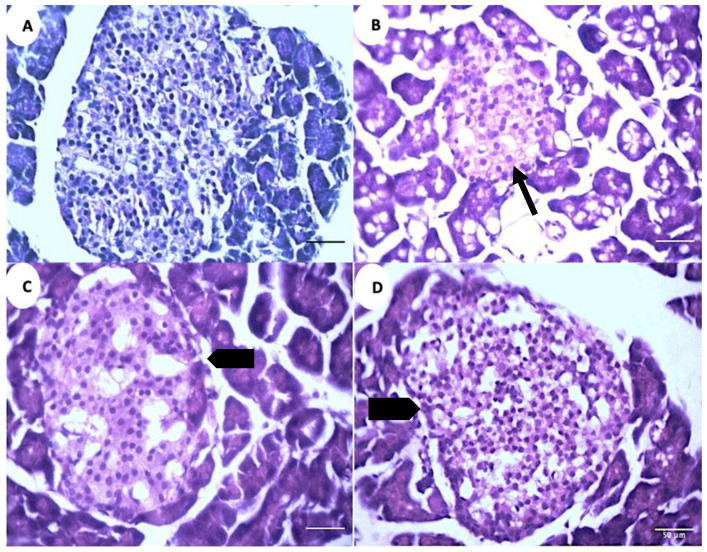
Representative photomicrographs of the islets of Langerhans in different treatment groups. (**A**). Negative control group (NC). (**B**). Positive control group (PC). (**C**). Treatment group 1 (GS1). (**D**). Treatment group 2 (GS2). Note the reduced diameter of the islet in the PC group (thin arrow) and the increase in the islet diameter in the GS2 treatment group (block arrow).

**Figure 9 metabolites-13-00516-f009:**
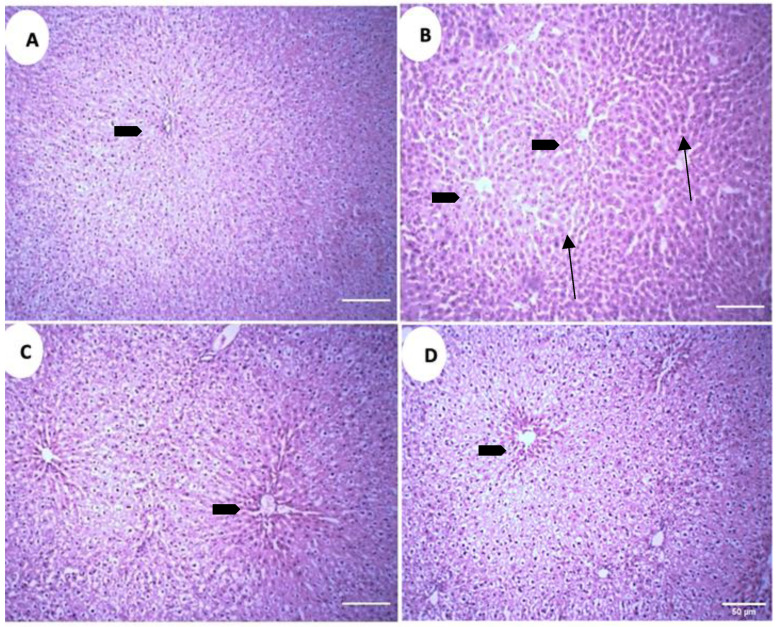
Representative photomicrographs of the liver parenchyma in different treatment groups. (**A**). Negative control group (NC). (**B**). Positive control group. (**C**). Treatment group 1 (GS1). (**D**). Treatment group 2 (GS2). Note the increase in the sinusoidal and perisinusoidal pace in the positive control group (PC) (arrows). Block arrow: central vein.

**Table 1 metabolites-13-00516-t001:** Detail of different groups and their subsequent treatment.

Groups	Treatments
Group I Normal (Negative control) n = 16	Receiving chow maintenance diet only(CMD)
Group II Diabetic (Positive control) n = 16	Receining CMD and pretreated with Alloxan(120 mg/kg b.w)
Group III Diabetic (Treatment group 1) n = 16	Receiving CMD + Supplemented with a dose of grinded powder of *G.sylvestre* 250 mg/kg BW in diet + Pretreated with Alloxan(120 mg/kg b.w)
Group IV Diabetic (Treatment group 2) n = 16	Receiving CMD + Supplemented with a dose of grinded powder of *G.sylvestre* 500 mg/kg BW in diet + Pretreated with Alloxan(120 mg/kg b.w)

## Data Availability

The data presented in this study are available within the article.

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
