# Peer review of "Gymnema Sylvestre Supplementation Restores Normoglycemia, Corrects Dyslipidemia, and Transcriptionally Modulates Pancreatic and Hepatic Gene Expression in Alloxan-Induced Hyperglycemic Rats"

_metabolites, 2023, doi:10.3390/metabo13040516_

Round 1
Reviewer 1 Report
The authors investigated a potential antioxidative, hepatoprotective, and hypocholesterolemic properties of a plant, Gymnema Sylvestre (GS), in an experimentally induced diabetic rat model. GS has been known for years in traditional medicine to help treat diabetes. I need to see the novelty of the subject covered. The manuscript is not prepared with sufficient care.
A group receiving the plant compound (GS) alone was not planned. What about potential toxicity? How were the applied doses selected?
It needs to be clarified what the letters a, b, c, and d above the bars mean.
Microscopic images of histological preparations do not include scale bars.
Author Response
I need to see the novelty of the subject covered.
The insulin-inducing ability of GS has been well established; however, the question of “does this increase in insulin is sufficient enough to induce hepatocyte signaling via insulin receptor substrates (Irs1, Irs2) and glucokinase (Gck) and modulate SREBP1c and ChREBP transcriptionally enough to corrects dyslipidemia?” is still unanswered. In the current study, we assessed the ability of insulin recovery by the GS treatment to correct dyslipidemia. Dyslipidemia is associated with either the lack of insulin and/or the absence of insulin signaling in the hepatocytes via Irs1, Irs2, Gck, and the subsequent transcription modulation of SREBP1c and ChREBP. We also explored the transcription modulation of Ins1 in the beta cells in the context of transcription factors like pdx1, MaFA, and Pax6, which reportedly enhance Ins1 transcription in the presence of increased plasma insulin concentration. The transcriptional control of hepatocytes was also explored regarding glucokinase (Gck) expression.
The manuscript is not prepared with sufficient care.
The manuscript has now been edited with a particular focus on language editing and according to the reviewer's comments.
A group receiving the plant compound (GS) alone was not planned. What about potential toxicity?
Previously in a 52-week dietary toxicity study, no-observed-effect level (NOAEL) has been documented for Gymnema Sylvestre leaves with a dose of 10,000mg/kg feed per day which corresponds to 504 mg/kg body weight per day for each animal (adult, male rats) (Ogawa et al., 2004). Also, Gymnema Sylvestre’s botanical preparations have been deemed safe by the German Federal Institute of Risk Assessment (BfR) with a lack of possible adverse effects. (Marakis et al., 2018)
Ogawa Y, Sekita K, Umemura T, Saito M, Ono A, Kawasaki Y, Uchida O, Matsushima Y, Inoue T and Kanno J, 2004. Gymnema sylvestre leaf extract: a 52-week dietary toxicity study in Wistar rats. Shokuhin Eiseigaku Zasshi, 45, 8–18.
German Federal Institute for Risk Assessment (BfR), Germany. Marakis, G, Ziegenhagen, R, Lampen, A and Hirsch-Ernst, KI, 2018. Risk assessment of substances used in food supplements: the example of the botanical Gymnema sylvestre. EFSA Journal 2018;16(S1):e16083, 10 pp.
How were the applied doses selected?
The dose rate for the animal trial was selected according to the previously published studies assessing the antidiabetic and anti-inflammatory potential of Gymnema Sylvestre supplementation in streptozotocin/alloxan-induced hyperglycemic adult rat models (Porchezhian et al., 2003; Khan et al., 2019).
Porchezhian E, Dobriyal RM. An overview on the advances of Gymnema sylvestre: chemistry, pharmacology and patents. Pharmazie. 2003 Jan;58(1):5-12.
Khan F, Sarker MMR, Ming LC, Mohamed IN, Zhao C, Sheikh BY, Tsong HF, Rashid MA. Comprehensive Review on Phytochemicals, Pharmacological and Clinical Potentials of Gymnema sylvestre. Front Pharmacol. 2019 Oct 29;10:1223.
It needs to be clarified what the letters a, b, c, and d above the bars mean.
Bars with similar letters are statistically non-significant. The clarification of the alphabetical letters on the bar graphs has been added to the figure legends.
Microscopic images of histological preparations do not include scale bars.
All the microscopic images of the histological preparations now include the scale bar.
Reviewer 2 Report
Reviewer comments and suggestions
The current study monitored the effect of Gymnema Sylvestre supplementation on beta cell and hepatic activity was explored in an alloxan-induced hyperglycemic rat model. Gymnema Sylvestre was supplemented @250mg/kg and 500mg/kg b.w. After 21 days, rats were euthanized, and blood and tissues (pancreas and liver) were collected for biochemical, expression, and histological analysis. The study result reported that group treated with Gymnema Sylvestre significantly reduced blood glucose levels with a subsequent increase in plasma insulin levels in a dosage-dependent manner. Total oxidant status (TOS), malondialdehyde, LDL, VLDL, ALT, AST, triglyceride, total cholesterol, total protein, C-reactive protein, and cortisol levels were reduced significantly in alloxan-treated hyperglycemic rats supplemented with Gymnema Sylvestre as compared to control. Significantly raised paraoxonase, arylesterase, albumin, and HDL levels were also observed in Gymnema Sylvestre-supplemented hyperglycemic rats. Different gene study was also done in pancreas and liver.
The current study indicates the potent effect of Gymnema Sylvestre which enhanced plasma insulin levels and further helped to improve hyperglycemia-induced dyslipidemia through transcriptional modulation of hepatocytes.
The paper objective was nice, and a few major modifications are needed to work on the manuscript.
- Line 35 please check the authenticity of the data
- Line 59 Please mention the approval number
- Line 99 Why it was done, please mention in one to two lines
- Line 132 purpose is to see what by histological analysis
- Line 233 enhanced word was not correct
- Section 4.7(title) of what, please complete it
- Figure 8 What does the arrow indicate, please mention it
- In lines 396-397 the authors can indicate figure or table here and other places as well
- Line 438-439 It Seems the sentence was incomplete, directly mention the conclusion before concluding other sentences
- All references need to be modified based on the MDPI Journals.
Author Response
1. Line 35 please check the authenticity of the data
The correction has been made regarding the data.
2. Line 59 Please mention the approval number
The approval number has now been mentioned.
3. Line 99 Why it was done, please mention in one to two lines.
The reason for assessing plasma levels of arylesterase activity has been mentioned in Line 98.
4. Line 132 purpose is to see what by histological analysis
The purpose of the histological analysis has been included in the text (Line 134).
5. Line 233 enhanced word was not correct
The word enhanced has been replaced by corrects (Line 236).
6. Section 4.7(title) of what, please complete it
The heading has been changed to “GS treatment significantly altered transcriptional profile in the pancreas and liver”. Section 3.6
7. Figure 8 What does the arrow indicate, please mention it
The arrow points towards the islet of Langerhans in the pancreatic tissue from different treatment groups.
8. In lines 396-397 the authors can indicate figure or table here and other places as well
Specific figures have now been indicated in the section.
9. Line 438-439 It Seems the sentence was incomplete, directly mention the conclusion before concluding other sentences
This section has been rewritten now according to the instruction in the above comment.
10. All references need to be modified based on the MDPI Journals.
All references are now modified based on MDPI journals.
Reviewer 3 Report
This is good paper with interesting findings. However, several flaws make this paper weaken as below.
1. Figure legends: General title of each figure should be written. do not start with (a) ...Also, legend style is really bad. Please check other papers and follow up good style to shortly explain experimental description and statistical information.
2. Authors have claimed that GS treatment improves pancreatic and liver tissue histology and then they also claimed there is transcriptional Modulation Of ßeta Cells And Hepatocytes in the title. Although authors have relevant results with tissue-specific genes, this reviewer does not feel it is correct. Since authors did not do any experiments with isloated beta cells and hepatocytes and rather they checked mRNA levels from the tissues. Histological approaches are still not giving detailed data targeting a specific cell. Therefore, to mention like above, authors should present experimental data observed with isolated beta cells or hepatocytes or their cell lines.
3. NC should be normal group and PC should be control. To make PC group, authors should have a group treated with known drug. Currently, no positive drug group is provided. Therefore, authors should include this to ensure their condition is good.
4. What type of nuclear factor kappa B (Nf-kB) genes were detected ? no supplementary information in current submission ! In addition, authors should present protein level results on the tissue-specific genes by Western blotting analysis to ensure the finding of mRNA levels.
Author Response
- Figure legends: General title of each figure should be written. do not start with (a) ...Also, legend style is really bad. Please check other papers and follow up good style to shortly explain experimental description and statistical information.
The figure legends have been changed according to the specific comments.
- Authors have claimed that GS treatment improves pancreatic and liver tissue histology and then they also claimed there is transcriptional Modulation Of ßeta Cells And Hepatocytes in the title. Although authors have relevant results with tissue-specific genes, this reviewer does not feel it is correct. Since authors did not do any experiments with isloated beta cells and hepatocytes and rather they checked mRNA levels from the tissues. Histological approaches are still not giving detailed data targeting a specific cell. Therefore, to mention like above, authors should present experimental data observed with isolated beta cells or hepatocytes or their cell lines.
The rightful specific concern raised in the above comment regarding the use of in-vitro models consisting of isolated beta cells and hepatocytes or cell lines to assess the antidiabetic potential of Gymnema Sylvestre. We would respond to this concern with; firstly, a handful of previously published studies exist employing in-vitro approaches like using isolated beta cells, beta cell lines, and isolated hepatocyte or hepatocyte-specific cell lines to understand the specific/mechanistic pathways through which Gymnema Sylvestre imparts its hypoglycemic and antidiabetic effect. Al-Romiyan et al. has been using different in-vitro approaches to understand the cellular response toward gymnema extracts and uncovered that increased Ca++ influx in isolated beta cells resulted in increased insulin secretion and enhanced beta cell survival via the activation of SOD and catalase enzyme system for protection against apoptosis (Al-Romiyan et al., 2010; Al-Romiyan et al., 2012; Al-Romiyan et al., 2013; Al-Romiyan et al., 2020). Secondly, the main objective of the current study was to understand the transcriptional status of key genes regulating glucose metabolism, lipid synthesis, and hepatic insulin signaling. As well as the transcriptional profile of the specific transcription factors involved in the regulation of the genes involved in the aforementioned roles as well as the insulin gene transcription in an in-vivo model. In the histological approach, our goal was to assess the size of islets of Langerhans and liver sinusoids instead of targeting specific cell type/types in the case of islets through immunocytochemistry.
Al-Romaiyan A, Liu B, Asare-Anane H, Maity CR, Chatterjee SK, Koley N, Biswas T, Chatterji AK, Huang GC, Amiel SA, Persaud SJ, Jones PM. A novel Gymnema sylvestre extract stimulates insulin secretion from human islets in vivo and in vitro. Phytother Res. 2010 Sep;24(9):1370-6.
Al-Romaiyan A, Liu B, Docherty R, Huang GC, Amiel S, Persaud SJ, Jones PM. Investigation of intracellular signaling cascades mediating stimulatory effect of a Gymnema sylvestre extract on insulin secretion from isolated mouse and human islets of Langerhans. Diabetes Obes Metab. 2012 Dec;14(12):1104-13.
Al-Romaiyan A, King AJ, Persaud SJ, Jones PM. A novel extract of Gymnema sylvestre improves glucose tolerance in vivo and stimulates insulin secretion and synthesis in vitro. Phytother Res. 2013 Jul;27(7):1006-11.
Al-Romaiyan, A, Liu, B, Persaud, S, Jones, P. A novel Gymnema sylvestre extract protects pancreatic beta-cells from cytokine-induced apoptosis. Phytotherapy Research. 2020; 34: 161– 172.
- NC should be normal group and PC should be control. To make PC group, authors should have a group treated with known drug. Currently, no positive drug group is provided. Therefore, authors should include this to ensure their condition is good.
The negative control group in the current experiment is indeed the normal group without any pretreatment or pathological condition. We agree with the reviewer that a known drug should have been used to generate the positive control group. Sulfonylureas are a known drug group that can enhance insulin secretion from beta cells and have been successfully used by humans for over 50 years. However, these excellent insulin secretagogues fail to mimic similar effects in rodent models of insulin deficiency and/or hyperglycemia (Niedowicz et al., 2018).
Niedowicz DM, Özcan S, Nelson PT. Glimepiride Administered in Chow Reversibly Impairs Glucose Tolerance in Mice. J Diabetes Res. 2018 Oct 29;2018:1251345.
- What type of nuclear factor kappa B (Nf-kB) genes were detected ? no supplementary information in current submission ! In addition, authors should present protein level results on the tissue-specific genes by Western blotting analysis to ensure the finding of mRNA levels.
The mRNA levels of the nuclear factor kappa B1 (Nf-kB1) gene were detected in the current study. The supplementary information has now been added to the current submission. The use of western blot to ensure the findings of mRNA levels is indeed a helpful tool to ensure protein levels, but the aim of our study was to examine the transcriptional profile. However, we agree with the reviewer's comments that assessing protein levels of transcriptionally active genes confirms the rt-PCR results, and we surely will be considering including western blot data to support our results in our future submissions to prestigious high-impact factor journals.
Reviewer 4 Report
The manuscript reports the effect of supplementation with Gymnema Sylvestre in Alloxan Induced Hyperglycemic Rats.
The work is interesting and can be a starting point for the creation of commercial products.
However, there are some points that should be clarified and improved in the manuscript:
- The title should be concise and shorter.
- The abstract should be improved and more specific.
- The introduction should be improved, including more topics covered in the manuscript.
- The authors used Gymnema Sylvestre leaves for the treatment. Its toxicity and composition must be reported in the manuscript. On the other hand, extraction of the active principles from the leaves could be carried out using water or ethanol. Why did the authors not use this option instead of using raw biomass (leaves)?
- Bibliographical references must be revised. For example, line 95, 100...
- Please review article structure, section numbering, add numbers to titles....
- Improve the quality of figures and tables.
- The authors identified in table 1 the treatment groups from I to IV. This must be maintained in figures and tables throughout the manuscript. In the captions, they should say what corresponds to each group.
- Statistical analysis should be improved. Compare results between treatment days for the same groups.
- Line 87-88: Total oxidant status (TOS) not total antioxidant status (TOS). Review and correct.
- Review and improve the expression of the units.
- The authors report that the treatments dose-dependently improve Arylesterase activity, MDA, ALT. In the results presented in figures 3 and 4 this is not observed. Please review this.
- Review figure 8. This figure was printed and the Grammarly program symbol appears in the figure. Pay attention to the formatting and use of images!
- In the microscopy captures, the scale must be included in each and every one of the figures. All of them are independent.
- In general, the expression and discussion of the results should be improved. Included evidence of the chemical composition and possible compounds involved in the action of this bioproduct.
Author Response
- The title should be concise and shorter.
The title has now been amended according to the reviewer's comment.
- The abstract should be improved and more specific.
The abstract has been improved with more specificity.
- The introduction should be improved, including more topics covered in the manuscript.
The introduction is now improved with more topics which are covered in the manuscript.
- The authors used Gymnema Sylvestre leaves for the treatment. Its toxicity and composition must be reported in the manuscript. On the other hand, extraction of the active principles from the leaves could be carried out using water or ethanol. Why did the authors not use this option instead of using raw biomass (leaves)?
Previously in a 52-week dietary toxicity study, no-observed-effect level (NOAEL) has been documented for Gymnema Sylvestre leaves with a dose of 10,000mg/kg feed per day which corresponds to 504 mg/kg body weight per day for each animal (adult, male rats) (Ogawa et al., 2004). Also, Gymnema Sylvestre’s botanical preparations have been deemed safe by the German Federal Institute of Risk Assessment (BfR) with a lack of possible adverse effects. (Marakis et al., 2018). Nearly all previously published studies assessing Gymnema Sylvestre’s antidiabetic and hypoglycemic effects used commercially available extracts or aqueous, ethanolic and methanolic extracts. However, the traditional use of gymnema plant as an antidiabetic in folk medicine has been in the form of dry leaf in powdered form.
Ogawa Y, Sekita K, Umemura T, Saito M, Ono A, Kawasaki Y, Uchida O, Matsushima Y, Inoue T and Kanno J, 2004. Gymnema sylvestre leaf extract: a 52-week dietary toxicity study in Wistar rats. Shokuhin Eiseigaku Zasshi, 45, 8–18.
German Federal Institute for Risk Assessment (BfR), Germany. Marakis, G, Ziegenhagen, R, Lampen, A and Hirsch-Ernst, KI, 2018. Risk assessment of substances used in food supplements: the example of the botanical Gymnema Sylvestre. EFSA Journal 2018;16(S1):e16083, 10 pp.
- Bibliographical references must be revised. For example, line 95, 100...
The correction has been made regarding the bibliography of the references e.g, line 95, 100
- Please review article structure, and section numbering, add numbers to titles....
The article structure has been reviewed and section and title numbering have been added.
- Improve the quality of figures and tables.
The quality of the figures and tables has been approved.
- The authors identified in table 1 the treatment groups from I to IV. This must be maintained in figures and tables throughout the manuscript. In the captions, they should say what corresponds to each group.
The description in table 1 about the treatment groups only corresponds to just mere numbers, and the experimental significance of each of these groups is defined by their designation e., negative control (NC) groups, positive control (PC), GS-treated group 1 (GS1), and GS-treated group 2 (GS2).
- Statistical analysis should be improved. Compare results between treatment days for the same groups.
Comparison of results between treatment days for the same group was only possible for the glucose levels as all the animals were sacrificed after 21 days, and biochemical, expression, and histological analysis was conducted on the blood and tissue samples collected on the day of sacrifice.
- Line 87-88: Total oxidant status (TOS) not total antioxidant status (TOS). Review and correct.
The correction has been made.
- Review and improve the expression of the units.
The expression of the units has been improved.
- The authors report that the treatments dose-dependently improve Arylesterase activity, MDA, ALT. In the results presented in figures 3 and 4 this is not observed. Please review this.
The results regarding Arylterase, MDA, ALT has been corrected.
- Review figure 8. This figure was printed and the Grammarly program symbol appears in the figure. Pay attention to the formatting and use of images!
Figure 8 has now been reviewed and formatted.
- In the microscopy captures, the scale must be included in each and every one of the figures. All of them are independent.
Scale bar has been included in each and every one of the figures.
Round 2
Reviewer 1 Report
The authors have satisfactorily responded to most of my questions and made the necessary changes to the manuscript. However, a few points have not been clarified:
It is still not very clear to mark statistical significance with the letters a, b, c, d without specifying towards which groups these values apply. And what about significance above 0.05, it is usually marked by multiplying the number of signs, the authors do not use this. Please mark statistical significance in a standard way (preferably *, #) and clearly describe in the legend to each figure against which group studied the given symbol means statistical significance and at what level (0.05 or 0.01 or 0.001).
I have doubts if the authors are entitled to use the specification "hepatocytes" in the title of the manuscript since the analyses were performed in the whole liver tissue
Author Response
1-Statistical significance has been added for every figure. The standard format as suggested by the reviewer has been used (*) and a clear description in the figure legend of each figure has been mentioned along with the appropriate level of statistical significance i.e, 0.05 or 0.01 or 0.001.
2- The title has been edited according to the reviewer's suggestion and the use of hepatocyte and beta cells has been avoided.
Reviewer 3 Report
Authors have fully addressed all issues. So, this paper is now acceptable.
Author Response
We thank the reviewers for their acceptance of the changes made by the authors on the reviewers suggestions.
Reviewer 4 Report
The authors made the changes taking into account the reviewers' comments. In this sense, the article has been substantially improved.
Minor comments:
- the expression of the units must be reviewed and standardized (pay attention to the entire manuscript).
- review the abbreviations. Use the abbreviation after the first use throughout the manuscript.
- the figures must be improved. There is a lot of manipulation, and inequality in the titles of the axes, text pasting, etc. Use a program to create images (eg GraphPad prism)
-
Author Response
- The expression of the units has been edited throughout the manuscript and now follows the same pattern of expression.
- The use of abbreviation has been corrected according to the reviewer's comments.
- Figures have been reformated according to the specific comments of the reviewers.